# GEOMETRIC ENHANCEMENT IN 3D GAUSSIAN SPLATTING FOR SPARSE-VIEW SCENE RECONSTRUCTION

## ABSTRACT

Although recent sparse-view scene reconstruction with 3D Gaussian Splatting (3DGS) like InstantSplat has made significant progress, it still suffers from geometric inconsistencies including floating artifacts, incomplete surface reconstruction, and unstable Gaussian primitives, which significantly degrade both visual quality and geometric fidelity. Additionally, the inaccurate camera pose will also exacerbate these issues. Therefore, we present a novel geometric enhancement framework for 3DGS including multi-view consistency enforcement and two geometric regularizations to fundamentally address these limitations. Specifically, our approach is composed of three key components: Side-view Inconsistency Filtering (SIF) at initialization, Local Depth Regularization (LDR), and Anisotropy-aware Shape Regularization (ASR) at training. The SIF module mainly leverages multi-view information to eliminate geometrically inconsistent points, which aims to reduce floating artifacts and improve surface coherence. LDR enforces spatial consistency by identifying and penalizing regions with high geometric uncertainty through patch-based depth correlation analysis. By controlling the opacity and scale ratio, ASR can constrain Gaussian primitives to geometrically plausible shapes, preventing degenerate elongated structures. Extensive experiments on two widely used datasets demonstrate the effectiveness and superiority of our geometric enhancement when compared to pose-free methods and even pose-known baselines.

## 1 INTRODUCTION

Reconstructing 3D scenes has always been a challenging task in computer vision and has been widely applied in virtual reality Kamran-Pishhesari et al. (2024), autonomous navigation Liao et al. (2025), and digital content creation Wang et al. (2023b). Based on Multi-View Stereo (MVS) Furukawa & Ponce (2009) and Structure-from-Motion (SfM) Schonberger & Frahm (2016), traditional methods require dense image collections and robust feature correspondences, and thus they will lead to incomplete reconstructions and significant geometric artifacts in scenarios with a limited number of viewpoints.

This naturally led to the emergence of sparse-view scene reconstruction. Recently, due to the application of Neural Radiance Fields (NeRFs) Mildenhall et al. (2021); Barron et al. (2022); Sitzmann et al. (2021) and Gaussian splatting (3DGS) Kerbl et al. (2023); Yan et al. (2024b); Feng et al. (2025); Zuo et al. (2025), significant progress has been made in sparse-view scene reconstruction, but it remains a formidable challenge. This difficulty stems from the inherently ill-posed nature of the problem: due to the limited observational constraints, the same set of 2D observations can correspond to multiple 3D structures, thereby causing inherent ambiguity in geometric estimation, which serves as an obstacle to achieving high-fidelity scene representations.

As illustrated in the top of Figure 1, floating artifacts are Gaussian primitives that appear in free space and can not correspond to actual surfaces, which is a particularly thorny issue. Such artifacts usually arise due to insufficient geometric constraints during the optimization process, causing the Gaussian primitives to deviate from the true geometry of the scene, thereby resulting in visually scattered elements to reduce the rendering quality. Another key challenge lies in the incomplete surface reconstruction, particularly in areas with limited coverage from input views, as evidenced by the bottom of Figure 1. Since sparse observations provide little information about occluded or

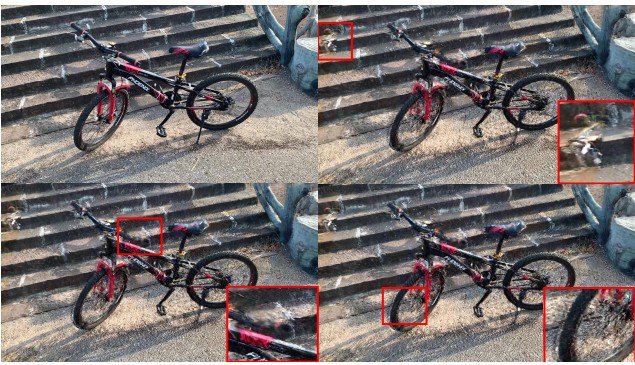

Figure 1: Visual examples of reconstruction challenges. The top images are: GT and floating artifacts, while the bottom images are both for incomplete surface reconstruction.

less-visible regions, the model may fail to place Gaussian primitives correctly, leading to holes or discontinuities in the reconstructed geometry. Furthermore, the emergence of unstable Gaussians with implausible shapes, such as overly elongated or flattened distributions, indicates that there are flaws in the underlying geometric representation. When camera poses are inaccurate, these problems will be further exacerbated, as misaligned viewpoints will introduce inconsistent spatial cues, often resulting in repetitive or distorted geometries.

Recent advances with various innovative strategies, including InstantSplat Fan et al. (2024) and MASt3R Leroy et al. (2024), have demonstrated the feasibility of sparse-view reconstruction via 3DGS techniques even without precise camera poses. However, these approaches often only address some individual aspects of the sparse-view problem, they do not provide a comprehensive solution to the geometric inconsistency for better sparse-view reconstruction. To this end, we propose a novel and comprehensive **G**eometric **E**nhancement framework in 3D **G**aussian **S**platting (dubbed **GEGS**) to systematically address these limitations, which is composed of three complementary modules. Specifically, Side-view Inconsistency Filtering (SIF) is a preprocessing strategy performed at initialization, which aims to identify and then remove those geometrically inconsistent points by leveraging multi-view depth and position consistency, to prevent error propagation and effectively mitigate floating artifacts from the outset. Then, during the optimization phase, Local Depth Regularization (LDR) is designed to enforce intra-patch depth coherence. Through analyzing viewpoint-dependent depth correlation within local neighborhoods, LDR penalizes regions with high uncertainty or inconsistency to enhance surface completeness in under-constrained areas. In the end, Anisotropy-aware Shape Regularization (ASR) constrains Gaussian primitives to remain within plausible geometric bounds by regulating scale anisotropy and opacity for effectively suppressing degenerate shapes that compromise both appearance and geometry.

In a word, we conclude our contributions as follows:

- We introduce a unified geometric enhancement framework with SIF, LDR, and ASR to improve reconstruction quality by eliminating geometry-ambiguous initializations and enforcing geometric coherences at training.
- Our GEGS strategy is compatible with existing pose-free 3DGS pipelines and can be seamlessly integrated into current frameworks without requiring additional supervision or pose refinement.
- Extensive experiments demonstrate that our method consistently outperforms existing pose-free approaches on multiple datasets by effectively preserving both visual fidelity and geometric consistency.

## 2 RELATED WORK

**Sparse-view Scene Reconstruction** aimed at recovering detailed geometry and appearance of a given scene from a limited number of input viewpoints. Prior works mainly focused on the regularization strategies of NeRF to solve the sparse-view challenge. For example, FreeNeRF Yang et al.

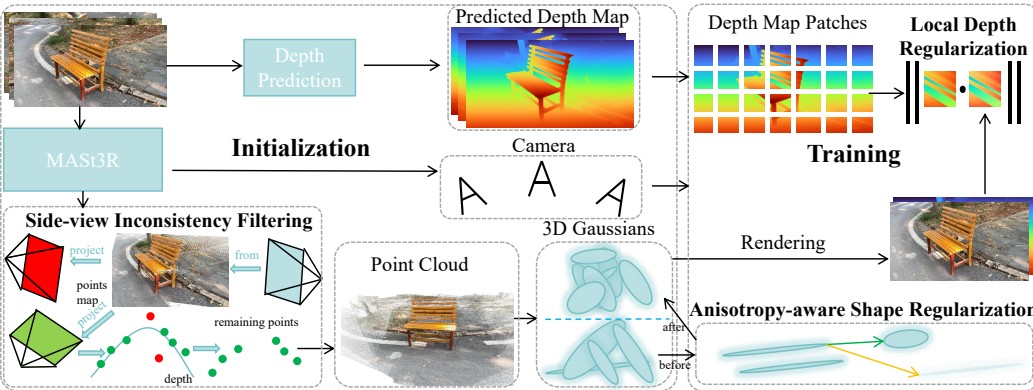

Figure 2: An overview of our GEGS pipeline. Following InstantSplat, we employ MASt3R to perform initialization, and then design SIF to remove redundant and erroneous points. At training, we jointly optimize with our novel regularizations: LDR enforcing local consistency, and ASR penalizing elongated Gaussians to improve geometric compactness.

(2023) employed frequency regularization to constrain the spectrum of learned radiance fields, and RegNeRF Niemeyer et al. (2022) combined with the constraints of depth smoothness and appearance consistency. The following methods like SparseNeRF Wang et al. (2023a), utilized multi-scale training with depth-guided sampling to improve convergence in sparse settings. Meanwhile, 3DGS brought new opportunities and challenges for sparse-view reconstruction. For instance, FSGS Zhu et al. (2024) demonstrated real-time few-shot view synthesis by incorporating specialized initialization strategies and adaptive densification on regions with sufficient observational support. DNGaussian Li et al. (2024) addressed scale ambiguity through global-local depth normalization, ensuring consistency between depth estimates and 3D Gaussian representations. CoR-GS Zhang et al. (2024) proposed a co-regularization framework that enforces consistency between multiple geometric representations, while DropGaussian Park et al. (2025) tackled structural regularization through strategic Gaussian elimination during training.

While they focus on individual geometric ambiguities, our GEGS strategy addresses the issue of geometric inconsistency from multiple perspectives both during initialization and training stages.

**Pose-free Neural Renderings.** The requirement for accurate camera poses significantly limits the applications of neural rendering. Despite the success in camera parameters estimation by traditional SfM methods such as COLMAP Schonberger & Frahm (2016), they often fail in sparse-view scenarios due to insufficient feature correspondences or challenging imaging conditions. Bundle-Adjusting Neural Radiance Fields (BARF) Lin et al. (2021) pioneered the pose-free reconstruction by jointly optimizing camera poses and neural radiance fields with a coarse-to-fine strategy that gradually increases the frequency components of positional encodings. On the shoulder of BARF, several subsequent works explored different aspects of pose-free neural rendering. GARF Chng et al. (2022) handled more challenging scenarios with larger pose uncertainties based on bundle adjustment. NoPe-NeRF Bian et al. (2023) incorporated monocular depth estimation as additional supervision to better estimate pose. CF-NeRF Yan et al. (2024a) progressively added new views to jointly optimized poses and scene representation. A concurrent work of CF-3DGS Fu et al. (2024) first attempted to eliminate the COLMAP dependency in Gaussian Splatting by jointly optimizing camera poses and Gaussian parameters, which demonstrated that the explicit nature of Gaussian representations can facilitate pose estimation through direct geometric alignment of Gaussian centers across frames. Recently, InstantSplat Fan et al. (2024) leverages a pre-trained multi-view stereo network by MASt3R Leroy et al. (2024) to initialize geometric estimates, and then achieves rapid sparse-view reconstruction even without SfM preprocessing. The method demonstrates significant speedup compared to traditional SfM+3DGS pipelines while maintaining competitive reconstruction quality.

Moreover, our GEGS comprehensively assures geometric consistency from several aspects to enhance the rendering quality under the pose-free condition.

## 3 METHOD

### 3.1 PRELIMINARY

Unlike NeRF Mildenhall et al. (2021), 3DGS Kerbl et al. (2023) explicitly represents a given scene as a collection of anisotropic 3D Gaussian primitives. Each primitive encodes both geometric and appearance attributes, and is directly optimized through differentiable rasterization. Each Gaussian primitive $G_i$ is parameterized by its 3D center $\boldsymbol{\mu}_i \in \mathbb{R}^3$, a positive semi-definite covariance matrix $\boldsymbol{\Sigma}_i \in \mathbb{R}^{3 \times 3}$, opacity $\alpha_i \in [0, 1]$, and view-dependent color coefficients represented via spherical harmonics (SH) Green (2003). The Gaussian density function is defined as:

$$G_i(\mathbf{x}) = \exp(-\frac{1}{2}(\mathbf{x} - \boldsymbol{\mu}_i)^\top \boldsymbol{\Sigma}_i^{-1}(\mathbf{x} - \boldsymbol{\mu}_i)). \tag{1}$$

To ensure numerical stability and positive semi-definiteness, the covariance matrix $\boldsymbol{\Sigma}_i$ is parameterized through eigendecomposition: $\boldsymbol{\Sigma}_i = \mathbf{R}_i \mathbf{S}_i \mathbf{S}_i^\top \mathbf{R}_i^\top$, where $\mathbf{R}_i \in SO(3)$ is a rotation matrix controlling the orientation of the Gaussian ellipsoid, and $\mathbf{S}_i = \text{diag}(s_i^x, s_i^y, s_i^z)$ is a diagonal scaling matrix with $s_i^x, s_i^y, s_i^z > 0$ representing the semi-axes lengths along the principal directions.

For differentiable rendering, 3D Gaussians are projected onto the image plane using the camera projection matrix $\mathbf{P} \in \mathbb{R}^{3 \times 4}$. The 2D covariance matrix $\boldsymbol{\Sigma}_i'$ for the projected Gaussian is computed as: $\boldsymbol{\Sigma}_i' = \mathbf{J} \mathbf{W} \boldsymbol{\Sigma}_i \mathbf{W}^\top \mathbf{J}^\top$, where $\mathbf{J}$ is the Jacobian of the projection transformation at $\boldsymbol{\mu}_i$, and $\mathbf{W}$ is the world-to-camera transformation matrix Hartley & Zisserman (2003). The final pixel color $C_p$ at pixel $p$ is computed through alpha compositing over all contributing Gaussians sorted in front-to-back order:

$$C_p = \sum_{i=1}^{N} c_i \alpha_i \prod_{j=1}^{i-1}(1 - \alpha_j), \tag{2}$$

where $c_i$ represents the color contribution of the $i$-th Gaussian, and $\alpha_i$ is its opacity after projection.

When the number of input views is severely limited, the reconstruction accuracy of 3DGS degrades significantly due to insufficient geometric constraints. Previous approaches Li et al. (2024); Zhang et al. (2024) have attempted to address this by incorporating global depth priors during training, typically formulated as:

$$\mathcal{L}_{3dgs} = \lambda_r \|\mathbf{C} - \hat{\mathbf{C}}\|_1 + \lambda_s \mathcal{L}_{\text{D-SSIM}}(\mathbf{C}, \hat{\mathbf{C}}) + \lambda_d \|d(\mathbf{D}, \hat{\mathbf{D}})\|_1, \tag{3}$$

where $\mathbf{C}$ and $\hat{\mathbf{C}}$ denote the ground truth and rendered images respectively, $\mathbf{D}$ and $\hat{\mathbf{D}}$ represent the corresponding depth maps, and $d(\cdot)$ is a depth consistency metric. Aforementioned, these methods typically require accurate camera poses by Structure-from-Motion (SfM) preprocessing Schonberger & Frahm (2016), which limits their applicability in practical scenarios only available with sparse and uncalibrated images.

### 3.2 OUR GEGS APPROACH

Based on the recent pose-free strategy InstantSplat Fan et al. (2024), we propose a novel geometric enhancement framework to systematically address the issue of geometric inconsistency in sparse-view 3DGS reconstruction. As illustrated in Figure 2, our GEGS strategy consists of three key components that operate at different stages in the reconstruction. Following InstantSplat, we first employ a pre-trained multi-view stereo network like MASt3R Leroy et al. (2024) to provide initial geometric estimations, which treats image matching as a 3D reconstruction problem by learning dense correspondences and 3D scene understanding simultaneously. Then, Side-view Inconsistency Filtering (SIF) is designed to identify and then remove those geometrically inconsistent initializations. During the optimization phase, Local Depth Regularization (LDR) will further penalize regions with high uncertainty or inconsistency to enhancing surface completeness in under-constrained areas. In the end, Anisotropy-aware Shape Regularization (ASR) constrains Gaussian primitives to remain within plausible geometric bounds.

**Side-view Inconsistency Filtering.** At initialization, MASt3R Leroy et al. (2024) may produce depth estimation errors, especially in areas with limited texture or ambiguous geometry Godard et al. (2019). These errors manifest as inconsistent depth predictions across different viewpoints,

Table 1: Comparison of average PSNR, SSIM, and LPIPS with different methods on the Tanks and Temples dataset. The best results for pose-free methods are highlighted in bold.

| | Method | PSNR↑ | | | SSIM↑ | | | LPIPS↓ | | |
|---|---|---|---|---|---|---|---|---|---|---|
| | | 12-view | 6-view | 3-view | 12-view | 6-view | 3-view | 12-view | 6-view | 3-view |
| Pose-known | COLMAP+3DGS | 30.01 | 25.33 | 18.24 | 0.917 | 0.808 | 0.601 | 0.095 | 0.171 | 0.348 |
| | COLMAP+FSGS | 30.17 | 25.70 | 19.88 | 0.914 | 0.814 | 0.638 | 0.099 | 0.169 | 0.302 |
| Pose-free | NoPe-NeRF | 17.22 | 15.56 | 14.89 | 0.582 | 0.496 | 0.444 | 0.411 | 0.516 | 0.587 |
| | CF-3DGS | 21.77 | 19.60 | 16.27 | 0.681 | 0.613 | 0.554 | 0.272 | 0.293 | 0.335 |
| | InstantSplat-XL | 28.51 | 25.35 | 23.59 | 0.883 | 0.849 | 0.753 | 0.106 | 0.121 | 0.188 |
| | **+ GEGS** | **30.22** | **27.86** | **23.70** | **0.926** | **0.872** | **0.764** | **0.093** | **0.100** | **0.174** |

which lead to floating artifacts and geometric instabilities if directly used for Gaussian initialization, evidenced by Figure 1. Thereby, SIF leverages cross-view geometric consistency to identify and remove erroneous points. For a given reference view $i$ with point cloud $\mathbf{P}_i = \{p_k^i\}$, we project each point into all other views $j \neq i$ and compare the projected depth with the corresponding depth prediction from MASt3R. Then, the side-view filtering for removing erroneous points from view $i$ can be formulated as:

$$\mathcal{M}_j = \begin{cases} 1 & \text{if } \left| \mathbf{D}_j \left( \pi_j \left( \mathbf{P}_i \right) \right) - \overline{\mathbf{D}}_j \left( \pi_j \left( \mathbf{P}_i \right) \right) \right| > \delta, \\ 0 & \text{otherwise}, \end{cases} \quad (4)$$

$$\mathbf{P}_i = (1 - \mathcal{M}_i) \cdot \mathbf{P}_i, \quad (5)$$

where $\pi_j(\cdot)$ denotes projection into view $j$, $\mathbf{D}_j$ is the reference depth in view $j$, and $\overline{\mathbf{D}}_j$ is the projected depth. The final mask $\mathcal{M}_i$ filters out points with large cross-view depth inconsistencies to reduce initialization noise and prevent error propagation during subsequent optimization.

**Local Depth Regularization**. Previous works Zhu et al. (2024) utilized global depth priors to successfully improve 3DGS, while often insufficient in capturing fine-grained details of complex scenes containing multiple objects. Inspired by DNGaussian Li et al. (2024), we propose a local depth regularization strategy to enforce consistency at a finer spatial granularity. Specifically, we first use a pre-trained monocular depth prediction model to estimate per-pixel depth map $\mathbf{D}^p$ for the training views. Then for each view $i$, we obtain the rendered depth map $\mathbf{D}^r$ from our current 3DGS model, and divide both the predicted and rendered depth maps into $K$ non-overlapping patches. Finally, we compute the Pearson correlation coefficient $\rho_k$ for each patch $k$, and define a global similarity threshold $\rho_{\text{global}}$ as the Pearson similarity over the full image. Patches with local similarity $\rho_k < \rho_{\text{global}}$ are considered inconsistent to be included in the local depth loss:

$$\mathcal{L}_{ldr} = \frac{1}{|\mathcal{S}|} \sum_{k \in \mathcal{S}} w_k \|\rho_k\|_1, \, \rho_k = \frac{Cov(\mathbf{D}_k^r, \mathbf{D}_k^p)}{\sqrt{Var_{\mathbf{D}_k^r} Var_{\mathbf{D}_k^p}}}, \quad (6)$$

where $\mathcal{S} = \{k \mid \rho_k < \rho_{\text{global}}\}$ is the set of selected inconsistent patches, and $w_k$ is a linearly assigned weight proportional to the severity of inconsistency (i.e., lower $\rho_k$ receives higher weight). This loss encourages the model to focus on local regions with the most significant errors, thereby improving the reconstruction quality of fine-grained details.

**Anisotropy-aware Shape Regularization.** In sparse-view scenarios, unconstrained optimization of 3D Gaussians often leads to degenerate shapes, particularly overly elongated Gaussians that span large spatial regions without corresponding to actual scene geometry Yu et al. (2024). These artifacts not only impair geometric interpretability but also lead to training instability and floating artifacts in novel view synthesis. To this end, ASR is designed to explicitly couple the opacity and shape characteristics of each Gaussian primitive. The key insight is that a Gaussian with extreme anisotropic shape should either reduce its opacity to minimize its visual impact or contract towards a more regular, isotropic form. Specifically, for each Gaussian $i$, we first compute its shape anisotropy ratio: $r_i = s_i^{\max}/s_i^{\min}$, where $s_i^{\max} = \max(s_i^x, s_i^y, s_i^z)$ and $s_i^{\min} = \min(s_i^x, s_i^y, s_i^z)$ represent the maximum and minimum scaling factors along the principal axes respectively.

We then define a shape-dependent penalty weight using a smooth activation function: $\omega_i = \sigma(\tau(r_i - T))$, where $\sigma(\cdot)$ is the sigmoid function, $\tau > 0$ is a temperature parameter to control the sharpness

Table 2: Comparison of average PSNR, SSIM, and LPIPS with different apporaches on the MVImgNet dataset. The best results for pose-free methods are highlighted in bold.

| | Method | PSNR↑ | | | SSIM↑ | | | LPIPS↓ | | |
|---|---|---|---|---|---|---|---|---|---|---|
| | | 12-view | 6-view | 3-view | 12-view | 6-view | 3-view | 12-view | 6-view | 3-view |
| Pose-known | COLMAP+3DGS | 23.11 | 18.98 | 14.72 | 0.712 | 0.618 | 0.313 | 0.216 | 0.390 | 0.529 |
| | COLMAP+FSGS | 24.31 | 21.66 | 16.68 | 0.798 | 0.706 | 0.544 | 0.215 | 0.309 | 0.462 |
| Pose-free | NoPe-NeRF | 16.41 | 15.88 | 14.82 | 0.492 | 0.463 | 0.447 | 0.423 | 0.517 | 0.588 |
| | CF-3DGS | 18.79 | 17.51 | 16.95 | 0.632 | 0.581 | 0.527 | 0.378 | 0.424 | 0.431 |
| | InstantSplat-XL | 23.51 | 21.78 | 18.11 | 0.738 | 0.685 | 0.563 | 0.241 | 0.276 | 0.349 |
| | **+ GEGS** | **24.62** | **22.51** | **19.20** | **0.778** | **0.698** | **0.583** | **0.216** | **0.260** | **0.334** |

of transition, and $T > 1$ defines the acceptable level of anisotropy. Thus, the shape regularization loss is formulated as:

$$\mathcal{L}_{asr} = \frac{1}{N} \sum_{i=1}^{N} \omega_i \cdot \alpha_i^2,$$ (7)

where $N$ is the total number of Gaussians and $\alpha_i$ is the opacity of the $i$-th Gaussian. This formulation creates a dynamic trade-off: the penalty weight $\omega_i$ increases as $r_i$ becomes large (indicating high anisotropy), which reversely encourages the optimizer to reduce $\alpha_i$ towards zero. Conversely, to maintain high opacity $\alpha_i$, the Gaussian must adopt a more balanced shape.

**Training Objective.** After SIF is performed at initialization, we combine the standard 3D Gaussian Splatting reconstruction loss $\mathcal{L}_{3dgs}$ with our proposed regularization terms, formulated as:

$$\mathcal{L} = \mathcal{L}_{3dgs} + \lambda_{ldr}\mathcal{L}_{ldr} + \lambda_{asr}\mathcal{L}_{asr},$$ (8)

where $\lambda_*$ are aimed at balancing the different contributions.

## 4 EXPERIMENTS

### 4.1 EXPERIMENTAL SETTINGS

**Datasets.** We conduct comprehensive experiments on two widely used multi-view datasets to thoroughly evaluate our effectiveness and generalization, as these datasets provide diverse scenarios with varying geometric complexity, lighting conditions, and scene types.

**Tanks and Temples Dataset** Knapitsch et al. (2017) consists of 8 challenging scenes: Ballroom, Bran, Church, Family, Francis, Horse, Ignatius, and Museum. The dataset features both indoor and outdoor environments with complex geometric structures and reflective surfaces. Each scene contains 150-300 images captured with calibrated cameras, and provides camera poses and intrinsic parameters.

**MVImgNet Dataset** Yu et al. (2023) includes 7 outdoor scenes featuring diverse object categories: Bench, Bicycle, Car, Chair, Ladder, SUV, and Table. MVImgNet provides a challenging benchmark with varying lighting conditions, object scales, and scene complexities. The images captured under natural lighting conditions. Unlike Tanks and Temples, MVImgNet focuses on object-centric scenes, allowing evaluation on different types of geometric structures.

**Evaluation Protocol.** To ensure fair and comprehensive evaluation under sparse-view settings, we adopt a systematic sampling strategy as InstantSplat Fan et al. (2024). For each scene, we uniformly sample 12 images as the testing set for novel view synthesis evaluation. From the remaining images, we further uniformly sample 3, 6, or 12 views as the training set to evaluate performance under different levels of view sparsity. Three standard metrics are used for quantitative evaluation. **Peak Signal-to-Noise Ratio (PSNR)** measures pixel-level reconstruction accuracy with higher values indicating better quality. **Structural Similarity Index Measure (SSIM) Wang et al. (2004)** evaluates perceptual similarity between rendered and ground truth images, considering luminance, contrast, and structure. **Learned Perceptual Image Patch Similarity (LPIPS) Zhang et al. (2018)** assesses perceptual quality using deep features, with lower values indicating better perceptual fidelity.

### 4.1.1 IMPLEMENTATION DETAILS

Our implementation is based on the PyTorch framework and all experiments are conducted on a single NVIDIA RTX 3090 GPU. In the initialization stage, we employ MASt3R with an input resolution of $512$, and utilize the cupy library to accelerate initialization. The threshold $\delta$ for Side-view Inconsistency Filtering is set to $0.1$. During the optimization stage, we train the model for $5000$ iterations. The patch size for LDR is set to $13 \times 13$ pixels, and predicted depth maps are obtained using Depth Anything V2 Large Yang et al. (2024), a monocular depth estimation model that provides reliable depth priors across diverse scene types. The patch weights are assigned by linear interpolation between $1.0$ and $2.0$. For the ASR, the temperature coefficient $\tau$ is set to $1.0$ and the anisotropy threshold $T$ is set to $5.0$. The base reconstruction loss $\mathcal{L}_{3dgs}$ follows the same configuration as previous works Zhu et al. (2024). The weight $\lambda_{ldr}$ for the LDR term is set to $0.3$, while the value of $\lambda_{asr}$ is set to $1.0$. For fairness, all other unspecified settings are consistent with InstantSplat Fan et al. (2024).

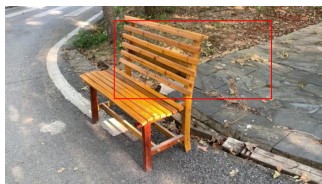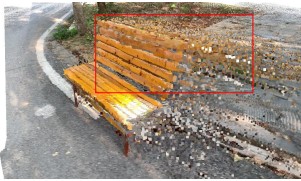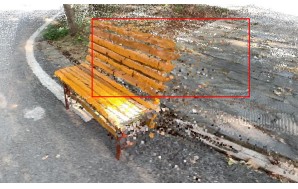

Figure 3: Visual comparison of initial point clouds with/out our SIF in the Bench scene, The images from left to right are GT, GT w/o SIF, GT with SIF, respectively.

## 4.2 EXPERIMENTAL RESULTS

We compare our GEGS method with several representative baselines published recently, including both pose-known and pose-free approaches. Specifically, 3DGS and FSGS are utilized by COLMAP to estimate accurate camera poses from the complete image set, and then sample sparse views as input for training. In contrast, Nope-NeRF, CF-3DGS, InstantSplat-XL, and our GEGS method are all pose-free methods that do not require to access accurrate camera poses. We report the detailed results in Table 1 and Table 2, respectively. A thorough analysis of these tables can easily lead to the following important conclusions:

1) For the dataset of Tanks and Temples, we can achieve better performance when compared with/ without the pose information across all different view inputting. Specifically, we can obtain an improvement up to 1.71dB at the metric of PSNR under 12-view input setting over the strongest baseline of InstantSplat-XL Fan et al. (2024). Similarly, performance enhancements were also observed in other metrics to varying degrees, such as from 0.883 to 0.926 in SSIM, and from 0.106 to 0.093 in LPIPS. Even under more challenging sparse view conditions of 6 or 3 inputs, our reconstruction quality can also improve remarkably, especially on the evaluations of SSIM and LPIPS.

2) On MVImgNet, the consistent and continuous performance uptrend can also be observed, with 1.11dB, 0.040, 0.025 improvements over InstantSplat-XL under 12-view conditions on PSNR, SSIM, LPIPS, respectively. Under the conditions of 6-view- and 3-view, the performance still showed a significant improvement, which indicates the effectiveness and superiority of our geometric enhancement framework in sparse viewpoint scene reconstruction.

3) Even compared with pose-known methods, our geometric enhancement without pose information renders comparable or even superior performance to pose-known baselines in almost scenarios. On Tanks and Temples with 12 views, our method (30.22dB PSNR) outperforms COLMAP+3DGS (30.01dB) and is marginally ahead of COLMAP+FSGS (30.17dB). This indicates that our geometric enhancement strategy can effectively compensate for the lack of accurate pose information by improving geometric consistency constraints.

In summary, the quantitative comparisons provided in Tables 1 and 2 comprehensively demonstrate that our GEGS strategy can win the pose-free SOTA method at different levels of view sparsity in all evaluation metrics. Even when compared with methods that require pose information, our GEGS

| CF-3DGS | InstantSplat-XL | Ours | Ground Truth |

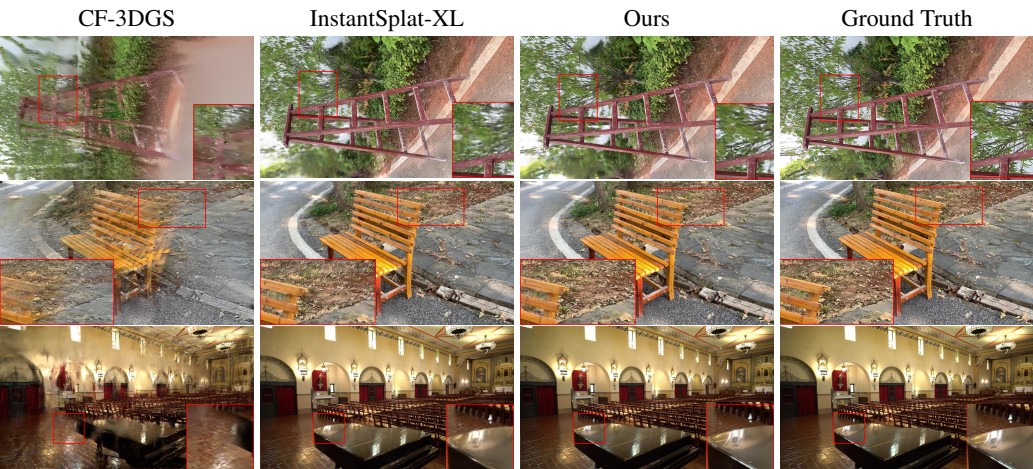

Figure 4: Visualized examples comparisons under 12-view input. Image sources, the first two:MVImgNet, and the third: T&T. The red boxes and their zoom in indicate that our reconstructions are with clearer geometric details and fewer artifacts.

still presents advanced performance, albeit to a lesser extent. All of these gains can be attributed to our geomentric consistnecy enhancements from different perspectives at initialization or training.

### 4.3 FURTHER ANALYSIS

**Ablation Studies** are solidly carried out on the MVImagNet dataset under 12-view input setting to investigate the effectiveness of each component in our GEGS framework from three aspects: (1) the impact of Side-view Inconsistency Filtering on the initialization quality and final reconstruction performance; (2) the contribution of Local Depth Regularization to the reconstruction quality; and (3) the effect of Shape Regularization on the final results.

Table 3: Ablation study of each component on the MVImgNet dataset under the 12-view setting.

| SIF | ASR | LDR | PSNR↑ | SSIM↑ | LPIPS↓ |
|---|---|---|---|---|---|
| ✗ | ✗ | ✗ | 23.51 | 0.738 | 0.241 |
| ✓ | ✗ | ✗ | 24.18 | 0.761 | 0.230 |
| ✗ | ✓ | ✗ | 23.89 | 0.752 | 0.235 |
| ✗ | ✗ | ✓ | 24.05 | 0.756 | 0.232 |
| ✓ | ✓ | ✗ | 24.32 | 0.769 | 0.222 |
| ✓ | ✗ | ✓ | 24.56 | 0.773 | 0.219 |
| ✓ | ✓ | ✓ | **24.62** | **0.778** | **0.216** |

We present the results of our ablation study in Table 3. When our SIF is integrated into the baseline, notable improvements on PSNR (+0.67dB), SSIM (+0.023) and LPIPS (-0.009) can be obtained, which indicates the importance of high-quality initialization for subsequent optimization. It can be further demonstrated by Figure 3, where SIF can obviously reduce noise and outliers in the initializations of point cloud. Benefiting from ASR's suppression on the degenerate of Gaussian primitives and its enhancement on geometric consistency, the continous combination will bring about further improvements by achieving at 24.32dB for PSNR. The better performance observed on the integration of LDR and SIF indicates the superiority of LDR to ASR in fine geometric alignment. Finally, the effective ensemble of the proposed three novel components can help the baseline to achieve the optimal performance, highlighting their complementary advantages in achieving a balance between initialization quality and geometric consistency.

**Parameter Analysis.** To better understand the effectiveness of our GEGS method, we continue to analyze the sensitivity of two weight coefficients $\lambda_{ldr}$ and $\lambda_{asr}$ for different geometric regu-

Table 4: Effect on weight of Local Depth Regularization (MVImgNet, 12-view input).

| $\lambda_{ldr}$ | MVImgNet 12-view | | |
|---|---|---|---|
| | PSNR↑ | SSIM↑ | LPIPS↓ |
| 0.01 | 24.36 | 0.769 | 0.215 |
| 0.02 | 24.54 | 0.772 | **0.209** |
| 0.03 | **24.56** | **0.774** | 0.214 |
| 0.05 | 24.28 | 0.767 | 0.231 |
| 0.07 | 24.33 | 0.771 | 0.211 |
| 0.10 | 24.35 | 0.770 | 0.212 |

Table 5: Effect on weight of Anisotropy-aware Shape Regularization (MVImgNet, 12-view input).

| $\lambda_{asr}$ | MVImgNet 12-view | | |
|---|---|---|---|
| | PSNR↑ | SSIM↑ | LPIPS↓ |
| 0.05 | 24.24 | 0.765 | 0.230 |
| 0.10 | 24.29 | 0.769 | **0.221** |
| 0.30 | 24.26 | 0.767 | 0.228 |
| 0.50 | 24.31 | 0.768 | 0.223 |
| 1.00 | **24.32** | **0.769** | 0.222 |
| 1.50 | 24.29 | 0.765 | 0.226 |

larizations, as displayed in Eq. 8. We set different numerical range for them according to their function,such as [0.01,0.1] for $\lambda_{ldr}$ and [0.05,1.5] for $\lambda_{asr}$, respectively. We fix the value of the selected parameter and search the best value of the left one, which could dynamically reflect our performance under different evaluation metrics.

From the detailed results in Table 4 and Table 5, we can observe that under all the metrics, the performance boosts as two coefficients rise up, and arrive at the peak when $\lambda_{ldr} = 0.03$ (24.56dB PSNR) and $\lambda_{asr} = 1.00$, respectively. Then the performance declines constantly when continuously increasing the values of the weights. Therefore, we obtain the optimals for different coefficients.

**Visualizations**. To better demonstrate our superiority on tackling geometric inconsistencies, we perform several visualizations of reconstructions results in MVImgNet under 12-view input between our GEGS method and two pose-known methods, one pose-free approach. As displayed by figure 4, our reconstruction quality are obviously better than the counterparts regardless of pose-known or pose-free strategy. For better understanding, we marked the local visuasizations in red rectangles. We attribute the improvement in reconstruction quality to the fact that our GEGS method can significantly reduce the geometric inconsistency.

## 5 CONCLUSION

In this paper, we presented a geometric enhancement framework in 3D Gaussian Splatting termed GEGS that addresses the challenges of sparse-view scene reconstruction without requiring pre-computed camera poses. Specifically, our method introduces three key components: Side-view Inconsistency Filtering for robust initialization, Local Depth Regularization for fine-grained geometric consistency, and Anisotropy-aware Shape Regularization for preventing degenerate Gaussian primitives. Extensive experiments on two widely used datasets demonstrate that our approach achieves a new bar of performance over existing pose-free methods with +1.71dB PSNR improvement under 12-view conditions and even larger gains under more challenging sparse-view scenarios. More importantly, our pose-free method can achieve comparable or even leading performance to pose-known baselines in several cases, highlighting the effectiveness of our geometric enhancements.

In future work, we plan to enhance the robustness under extreme sparse-view conditions by incorporating semantic priors or large-scale vision foundation models.

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
