# Supplementary Material for Geometric Enhancement in 3D Gaussian Splatting for Sparse-view Scene Reconstruction

## Abstract

This supplementary material demonstrates the effectiveness of the Side-view Inconsistency Filtering (SIF) method in 3D point clouds initialization through experimental evaluations on the Tanks and Temples datasets.

## 1 Supplementary Content

### 1.1 Additional Experiments

#### 1.1.1 Initialization Quality Evaluation

To validate the effectiveness of our Side-view Inconsistency Filtering (SIF) component, we conduct a direct evaluation of initialization quality on the Tanks and Temples dataset under 12-view settings. We project the initial 3D point clouds to all viewpoints and compare the rendered images with ground truth using standard metrics. As shown in Table 1 and Figure 1, SIF significantly improves the initialization quality by filtering out geometrically inconsistent points, achieving 0.46dB PSNR improvement and 0.006 SSIM enhancement while reducing the point cloud size by 12.2% (from 1,187,258 to 1,042,437 points). This demonstrates that SIF effectively removes noisy initializations and provides a cleaner geometric foundation for subsequent optimization, validating our multi-view consistency filtering strategy.

Table 1: Initialization Quality Comparison on Tanks and Temples (12-view)

| Method | PSNR↑ | SSIM↑ | LPIPS↓ | Point Count |
|---|---|---|---|---|
| w/o SIF | 18.64 | 0.518 | 0.346 | 1,187,258 |
| w/ SIF | **19.10** | **0.524** | 0.350 | 1,042,437 |
| Improvement | +0.46 | +0.006 | +0.004 | -144,821 |

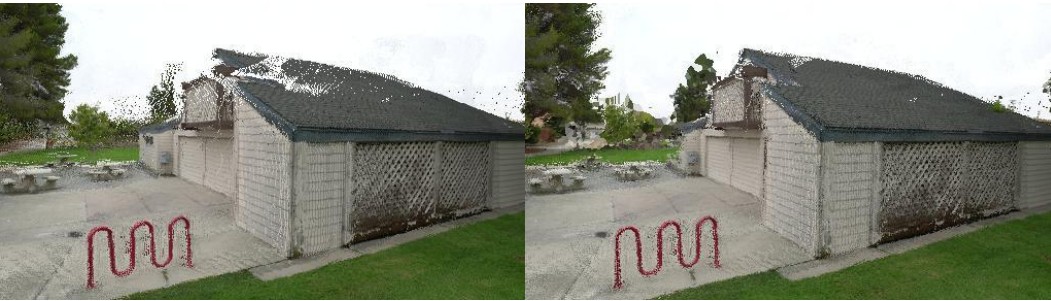

Figure 1: Visual comparison of initialization quality without (left) and with (right) SIF on representative scenes from Tanks and Temples dataset. Our SIF effectively removes noisy and inconsistent points while preserving geometric structures.

## References