# OpenReview forum: "Geometric Enhancement in 3D Gaussian Splatting for Sparse-view Scene Reconstruction"
_ICLR.cc/2026/Conference — ICLR 2026 Conference Withdrawn Submission_

### Official Review · Reviewer_WG54 · 2025-10-25

**Soundness:** 3
**Presentation:** 3
**Contribution:** 2
**Rating:** 4
**Confidence:** 4

**Summary:**

This paper addresses pose-free sparse-view 3D reconstruction with Gaussian splatting, where the lack of camera poses and limited views cause floating artifacts, missing surfaces, and shape degeneracy. The authors propose GEGS, a plug-in framework with three components: (1) a side-view inconsistency filter that removes geometrically self-contradictory points at initialization, (2) a local depth regularization that focuses supervision on regions with high depth mismatch, and (3) an anisotropy-aware regularizer that suppresses implausibly elongated Gaussians via opacity coupling. Integrated into a pose-agnostic 3DGS pipeline, GEGS consistently improves reconstruction quality across 3/6/12-view settings on standard benchmarks.

**Strengths:**

* The paper proposes a strategy for pose-free sparse-view 3DGS by combining initialization filtering, depth-guided local correction, and shape-aware suppression in a principled way rather than ad-hoc fixes.
* The methodology is well-justified by the identified failure modes, implemented cleanly, and validated through ablations.
* The contribution is practically relevant, expanding the applicability of 3D Gaussian methods to more realistic acquisition scenarios.

**Weaknesses:**

* Although the problem is framed as “pose-free,” the method still consumes MASt3R to derive an initial geometry, which itself implicitly uses multi-view structure cues; the paper does not compare against explicit pose estimation baselines to clarify whether accuracy is gained or merely offloads the pose problem to MASt3R.
* SIF, LDR assumes monocular depth is a sufficiently reliable prior for sparse-view correction, but in real unconstrained scenes monocular depth can be systematically wrong; the paper lacks a failure or robustness analysis in such cases.
* ASR suppresses elongated Gaussians indiscriminately; some legitimate structures (e.g., lamp posts, thin edges) are naturally anisotropic, yet the paper does not examine whether ASR over-smooths or erases these classes of geometry.

**Questions:**

Please refer to the weaknesses section above — all of my questions directly correspond to the issues discussed there.

---

### Official Review · Reviewer_z3Kk · 2025-10-30

**Soundness:** 2
**Presentation:** 3
**Contribution:** 2
**Rating:** 4
**Confidence:** 4

**Summary:**

This paper presents GEGS (Geometric Enhancement in 3D Gaussian Splatting), a framework for improving sparse-view scene reconstruction without relying on camera pose supervision. The authors identify key limitations in existing pose-free 3DGS pipelines such as geometric inconsistency, floating artifacts, and degenerate Gaussian shapes, and propose three complementary modules to address these issues: 1) Side-view Inconsistency Filtering (SIF) — a preprocessing step that removes geometrically inconsistent points via cross-view depth comparison. 2) Local Depth Regularization (LDR) — a training-time regularization that enhances local geometric coherence by penalizing regions with low depth correlation. 3) Anisotropy-aware Shape Regularization (ASR) — a geometric prior linking Gaussian opacity and shape anisotropy to prevent elongated, implausible primitives. Extensive experiments on Tanks and Temples and MVImgNet demonstrate that GEGS outperforms the pose-free baseline InstantSplat, and in some cases achieves comparable or superior results to pose-known methods such as COLMAP+3DGS and FSGS. The framework is modular, computationally efficient, and compatible with existing 3DGS pipelines.

**Strengths:**

1. The paper introduces a fresh way of tackling common problems in 3D Gaussian Splatting (3DGS) for sparse-view reconstruction. By using techniques like Side-view Inconsistency Filtering (SIF), Local Depth Regularization (LDR), and Anisotropy-aware Shape Regularization (ASR), the authors improve the quality of 3D reconstructions by reducing issues like floating artifacts and incomplete surfaces.
2. The method is designed to fit easily into current 3DGS frameworks. This means it can be adopted without needing major changes to existing systems, making it practical for researchers and developers in the field.
3. The paper is written in a straightforward and well-structured way, making it easy for readers to follow the logic behind the proposed methods.

**Weaknesses:**

1. It has limited novelty. Each of the three modules (SIF, LDR, ASR) builds upon well-established ideas: multi-view geometric filtering, local depth correlation, and anisotropy regularization.  While the combination is practical and empirically beneficial, the individual techniques are incremental rather than fundamentally new.
2. The idea of ASR lacks detailed theoretical justification. The motivation behind coupling opacity and anisotropy in ASR is intuitive but lacks rigorous analysis. The choice of sigmoid weighting and hyperparameter settings (e.g., τ, T) appears empirical, with little discussion of their geometric interpretation or stability.
3. The evaluation scope is narrow. Although the paper presents results on two datasets, both are relatively standard for 3DGS evaluation. Additional experiments on real-world uncalibrated datasets (e.g., LLFF or ScanNet subsets) could better validate pose-free robustness.
4. While Table 3 shows that combining all modules yields the best results, the analysis does not clarify why LDR contributes more than ASR or whether SIF occasionally discards useful geometry. Quantitative evidence on how each component alters point density or Gaussian stability would strengthen the claims.
5. The method adds several regularization terms and filtering steps. However, no runtime, memory, or convergence analysis is provided to demonstrate that the geometric enhancements do not significantly slow down optimization compared to InstantSplat.

**Questions:**

1. Role of SIF in initialization: The paper claims that SIF removes geometrically inconsistent points, but how is δ = 0.1 chosen? Could overly aggressive filtering remove correct but occluded structures? A quantitative measure of remaining point density before and after SIF would help.
2. Since LDR depends on monocular depth priors (Depth Anything V2), how sensitive is the method to inaccurate depth estimation? Have the authors tried training without any external depth predictor to evaluate robustness?
3. The opacity–anisotropy coupling seems heuristic. Could you provide visualizations of how αi evolves for Gaussians with high anisotropy ratio during training? Does this penalty risk over-smoothing fine details?
4. What is the added training time per iteration due to LDR and ASR compared with InstantSplat? Are the methods feasible for real-time or interactive reconstruction tasks?

---

### Official Review · Reviewer_1oPW · 2025-11-01

**Soundness:** 2
**Presentation:** 3
**Contribution:** 2
**Rating:** 4
**Confidence:** 4

**Summary:**

This paper improves sparse-view 3DGS through geometric consistency and regularizations. The method first utilizes MASt3R to estimate camera parameters and point clouds, and leverages cross-view geometric consistency to filter inconsistent points for 3DGS initialization. Then, leveraging monocular depth estimation, the method uses NCC to impose local depth regularization. Moreover, the method proposes anisotropy-aware shape regularization based on the opacity and the scale ratio to alleviate overly entangle Gaussians. Experiments show that the method effectively improve InstantSplat-XL.

**Strengths:**

1. The approach makes full use of geometric consistency and regularization strategies to improve sparse-view 3DGS performance..
2. Experimental results demonstrate that the approach further boosts the performance of InstantSplat-XL.
3. The paper is well written and easy-to-follow.

**Weaknesses:**

1. The geometry regularization is widely used in sparse-view reconstruction. This makes the novelty of the method limited. There exist similar operations in other works. For local depth regularization, it is better to compare it with the local depth in DNGaussian. For anisotropy-aware shape regularization, it is better to compare it with the rank constraint in [a].
[a] Hyung et al, Effective Rank Analysis and Regularization for Enhanced 3D Gaussian Splatting, NeurIPS 2024.
2. The experimental results are limited. There exist many SOTA methods for sparse-view 3DGS, such as DNGaussian, Cor-GS, DropGaussian and Nexusgs [b]. It is better to compare quantitative results with these methods.
[b] Zheng et al. Nexusgs: Sparse view synthesis with epipolar depth priors in 3d gaussian splatting, CVPR 2025.
3. The work focuses on geometry enhancement in 3DGS, however, no relevant experiments show that the method improves depth map estimations.

**Questions:**

1. Can you compare the training efficiency of different methods?
2. It is better to conduct experiments on widely used MipNeRF-360, DTU and LLFF datasets.
3. Can you show if the method improves depth estimation?

---

### Official Review · Reviewer_PBUR · 2025-11-01

**Soundness:** 2
**Presentation:** 2
**Contribution:** 2
**Rating:** 2
**Confidence:** 4

**Summary:**

The paper tries to address the sparse-vew synethsis problem with 3D Gaussian Splatting (3DGS). The proposed method leverages the Mast3r initialized point cloud then introduces Side-view Inconsistency Filtering (SIF) to remove geometrically incorrect points. Local Depth Regularization (LDR) and Anisotropy-aware Shape Regularization (ASR) are proposed to enforce spatial consistency and constrain the Gaussian primative shapes.

**Strengths:**

1. The paper is overally well-written and easy to follow.
2. The proposed method outperforms Pose-free and Pose-known methods based on the quantitative results.
3. The SIF module makes sense and recudes the artifacts.

**Weaknesses:**

1. The qualitative comparision with other methods are hard to see in Figure 4. from my perspective.
2. The paper lacks references: SPARS3R[1], which also investigates the Mast3r point cloud as initialization, and [2], which is a pose-free sparse-view synthesis method.
3. The LDR and ASR are similar in DNGaussian[3] and PhysGaussian[4].

[1]. Tang, Y., Guo, Y., Li, D., & Peng, C. (2025). SPARS3R: Semantic Prior Alignment and Regularization for Sparse 3D Reconstruction. In Proceedings of the Computer Vision and Pattern Recognition Conference (pp. 26810-26821).

[2]. Jiang, K., Fu, Y., Varma T, M., Belhe, Y., Wang, X., Su, H., & Ramamoorthi, R. (2024, July). A construct-optimize approach to sparse view synthesis without camera pose. In ACM SIGGRAPH 2024 Conference Papers (pp. 1-11).

[3]. Li, J., Zhang, J., Bai, X., Zheng, J., Ning, X., Zhou, J., & Gu, L. (2024). Dngaussian: Optimizing sparse-view 3d gaussian radiance fields with global-local depth normalization. In Proceedings of the IEEE/CVF conference on computer vision and pattern recognition (pp. 20775-20785).

[4] Xie, T., Zong, Z., Qiu, Y., Li, X., Feng, Y., Yang, Y., & Jiang, C. (2024). Physgaussian: Physics-integrated 3d gaussians for generative dynamics. In Proceedings of the IEEE/CVF Conference on Computer Vision and Pattern Recognition (pp. 4389-4398).

**Questions:**

1. What are the limitations of the proposed method?

---

### Note · Authors · 2025-12-10

I have read and agree with the venue's withdrawal policy on behalf of myself and my co-authors.